# Rapid and Sensitive Electrochemical Assay of Cefditoren with MWCNT/Chitosan NCs/Fe_2_O_3_ as a Nanosensor

**DOI:** 10.3390/mi13081348

**Published:** 2022-08-19

**Authors:** Nida Aydogdu, Goksu Ozcelikay, Sibel A. Ozkan

**Affiliations:** 1Department of Analytical Chemistry, Faculty of Pharmacy, Afyonkarahisar Health Sciences University, Afyonkarahisar 03218, Turkey; 2Department of Analytical Chemistry, Faculty of Pharmacy, Ankara University, Ankara 06560, Turkey

**Keywords:** cefditoren, cephalosporins, antibiotics, β-lactamases, determination, electroanalytical methods, electrochemical, voltammetry

## Abstract

In this research, a glassy carbon electrode (GCE) modified by MWCNT/chitosan NCs/Fe_2_O_3_ was prepared for the determination of the cephalosporin antibiotic cefditoren (CFT) using adsorptive stripping differential pulse and cyclic voltammetry techniques. The effects of pH, the scan rate, the deposition potential, the accumulation time, and modification agents on the determination of CFT were analyzed. The results showed that the modified electrode significantly increased the oxidation peak current of CFT. Under optimized conditions, the MWCNT/chitosan NCs/Fe_2_O_3_/GCE nanosensor exhibited a linear response between 0.2 µM and 10 µM toward CFT. The limit of detection and quantification were determined to be 1.65 nM and 5.50 nM, respectively. Model drugs (cefdinir, cefpodoxime, cephalexin, and ceftazidime compounds) were used to enlighten the CFT oxidation mechanism. Moreover, the nanosensor was used to analyze CFT in a pharmaceutical dosage form and commercial deproteinated human serum samples. The accuracy of these methods was proven in the recovery studies, with values of 96.98 and 98.62% for the pharmaceutical dosage form and commercial deproteinated human serum sample, respectively.

## 1. Introduction

Cefditoren (CFT) is a third-generation cephalosporin antibiotic. This drug is used for the treatment of biotic disorders, including skin infections, palatine tonsils, and acute and chronic bronchitis. It shows antibacterial activities for Gram-negative and Gram-positive pathogens. The dosage of CFT can be 100, 200, and 400 mg. The C_max_, t_max_, and half-life are 3.4 mcg/L, 2.0 h, and 1.1 h, respectively. The rapid detection of antibiotics can contribute to antibiotic treatments with favorable clinical effects and a decrease in inappropriate antibiotic usage [1]. The structure of CFT, including the cephem skeleton, is shown in Figure 1.

Several analytical techniques have been performed for the determination of cefditoren, including LC-MS techniques [2], voltammetrics [3], spectrophotometry [4,5,6], chemiluminescence [7], micellar liquid chromatography [8], high-performance liquid chromatography (HPLC) [5,9,10,11,12,13], Fourier transform infrared spectroscopy [14], potentiometry using ion-selective electrodes [15], and thin-layer chromatography (TLC) [5]. Electrochemical techniques are significant for monitoring and sensing pharmaceuticals because of their specific properties such as a fast response, sensitivity, no pre-treatment, simplicity, low organic solvent consumption, and low cost versus traditional analysis methods [16]. Voltammetric techniques, including squarewave voltammetry (SWV) and differential pulse voltammetry (DPV), have been used to determine the electroactive pharmaceutical compounds in different samples. Cyclic voltammetry (CV) has been applied to investigate the oxidation and reduction mechanisms of drug compounds [17]. It has been noted that the AdSDPV technique is more sensitive to the current obtained by the oxidation of the compound deposited on the surface by potential scanning than the DPV technique. The AdSDPV technique has a high sensitivity and selectivity.

Moreover, the electrochemical sensor performance can be enhanced by the interaction between the analyte and electrode surface by the modification of different nanomaterials. Carbonaceous and metal nanoparticles serving as electrode-surface modifiers have many advantages such as a wide potential window, a high surface-to-volume ratio, a low detection limit, easy functionalization, and a low background current. The combined use of such nanomaterials creates synergistic effects [18,19]. A nanocomposite exhibiting synergism has been proposed to demonstrate a multi-fold enhancement in the sensing ability of an electrode. Multi-walled carbon nanotubes (MWCNTs) are composed of multiple single-wall carbon nanotubes weakly bound by van der Waals interactions [20]. Magnetic materials have significance because of their excellent physicochemical properties such as their electrochemical properties, widespread availability, low synthesis expense, and non-toxicity. Iron III oxide (Fe_2_O_3_) nanomaterials are one of the magnetic metal oxide nanoparticles ranging from 1–100 nm. Fe_2_O_3_ is preferred due to its high surface area/volume ratio, good biocompatibility, low toxicity, strong magnetic properties, easy preparation, and low cost [21]. Chitosan is a cross-linker hydrophilic polymer. It has many application fields because of its excellent properties such as a “smart” nanostructure and it is electrical, catalytic, reusable, and multi-functional [22].

In this work, an MWCNT/chitosan NCs/Fe_2_O_3_-based electrochemical sensor was successfully developed for the sensitive assay of CFT. The electrochemical performance of CFT was evaluated by different electrochemical methods (CV, DPV, and AdSDPV). MWCNT/chitosan NCs/Fe_2_O_3_ significantly improved the signals compared with a bare GCE. Based on the physicochemical and electrochemical properties of the nanocomposite, a novel simple nanosensor was also constructed for CFT detection. Benefiting from the good physicochemical and electrochemical properties of the nanocomposite, this nanosensor showed a high sensitivity, high accuracy, excellent reproducibility, and a wide linear range for CFT as well as good performance in the spike sample detection.

## 2. Materials and Methods

### 2.1. Chemicals and Reagents

In the present work, all reagents used were of an analytical standard and were used without pre-processing. CFT was supplied by Sigma-Aldrich. Spectracef^®^ pediatric sachets, the pharmaceutical dosage form, were obtained from a topical pharmacy. Drug-free human serum from male AB plasma was received from Sigma-Aldrich. Sodium acetate trihydrate (>99%), acetic acid (>99%), phosphoric acid (>85%), acetonitrile (99.8%), methanol (99.8%), sodium dihydrogen phosphate dihydrate (>99%), sodium phosphate monobasic (≥99.0%), sodium phosphate (≥99.0%), sulfuric acid (95–97%), sodium hydroxide (>97%), N, N-dimethyl formamide (DMF), boric acid (>99%), hydrochloric acid (37%), potassium chloride (KCl), magnesium sulphate (MgSO_4_), sodium nitrate (NaNO_3_), ascorbic acid, glucose, dopamine, uric acid, and paracetamol were procured from Sigma-Aldrich. Fe_2_O_3_, chitosan NCs, and the MWCNTs were obtained from Sigma-Aldrich. 

### 2.2. Apparatus

For the electrochemical experiments, differential pulse voltammetry (DPV), adsorptive stripping differential pulse voltammetry (AdSDPV), and cyclic voltammetry (CV) were performed by a PalmSens Potensiostat (Utrecht, The Netherlands) electrochemical analyzer running with PSTrace 5.8 software. The three-electrode cell system comprised a glassy carbon electrode (GCE) with a diameter of 3.0 mm as a working electrode, an Ag/AgCl electrode (3 M KCl) as the reference electrode, and Pt wire as the counter electrode. These were used for the voltammetric analysis of CFT. An electronic balance (Ohaus, Parsippany-Troy Hills, NJ, USA) was used for the precise weighing of the chemicals. A ZEISS EVO 40 (Merlin, Carl Zeiss, Weimar, Germany) was used to acquire the EDX spectra and scanning electron microscopy (SEM) images. A pH meter (Mettler-Toledo pH/ion S220, Greifensee, Switzerland) with an InLab Expert Pro-ISM glass electrode was used for the pH measurement of a buffer solution with an accuracy of ±0.05.

### 2.3. Optimization Procedures

The AdSDPV studies were performed at optimum parameters of a 0.0 s equilibration time, 0.005 V step potential, 0.02 V pulse potential, 0.07 s pulse time, and 0.02 V s^−1^ scan rate for the MWCNT/chitosan NCs/Fe_2_O_3_/GCE nanosensor. In the AdSDPV technique, the optimum stripping conditions were 0.0 V for the deposition potential and 150 s for the accumulation time. Cyclic voltammograms were saved at a scan rate ranging from 0.005 to 1.0 V s^−1^ to understand the CFT mechanism. All voltammetric measurements were taken at room temperature (25 °C).

### 2.4. Preparation of the Modified Electrodes

At the beginning of the analysis, the bare GCEs were cleaned with ethanol:distilled water (1:1) in an ultrasonic bath for 15 min. The working electrode surface was then polished with polycrystalline alumina slurry and washed out with distilled water. Fe_2_O_3_ (1 mg mL^−1^) and MWCNTs (1 mg mL^−1^) were dispersed in DMF. Chitosan (0.1%, *w*/*v*) was prepared in an acetate buffer (pH 4.7). For the modification of the GCE, Fe_2_O_3_, chitosan, and MWCNTs were mixed in a 1:1:1 (*v*/*v*/*v*) ratio. An amount of 1 μL of this mixture was enforced on the bare GCE and dried under a vacuum for 15 min. Before the measurements, the modified electrode surface was regenerated 5 times with a CV scan via the electrochemical technique. The voltammograms of the prepared MWCNT/chitosan NCs/Fe_2_O_3_/GCE nanosensors were saved.

### 2.5. Preparation of the Solutions

A total of 1.0 mM CFT stock solution was prepared by dissolving in acetonitrile. The standard solutions used for the calibration plot were prepared by diluting the stock solution with a supporting electrolyte containing acetonitrile concentrations (kept to a constant 20% (*v*/*v*)). Different supporting electrolyte solutions such as sulfuric acid (pH 0.3–1.0), a Britton–Robinson (BR) buffer (pH 2.0–12.0), a phosphate buffer (pH 1.5–8.0), and an acetate buffer (pH 3.7–5.7) used with distilled water in the electrochemical analysis were prepared. DMF was used as the solvent of the MWCNTs and Fe_2_O_3_ (1.0 mg mL^−1^) and an acetate buffer at pH 4.7 was used to prepare the chitosan (0.1 mg mL^−1^). These prepared solutions were mixed in an ultrasonic bath for 2 h. The nanocomposite was achieved by mixing the solutions at a ratio of 1:1:1 (*v*/*v*/*v*). Stock solutions of interferents such as dopamine, glucose, uric acid, ascorbic acid, and inorganic compounds such as 1.0 mM MgSO_4_, 1.0 mM KCl, and 1.0 mM NaNO_3_ used in interference studies were prepared in distilled water. All supporting electrolytes and the standard stock were stored in a fridge at +4 °C.

### 2.6. Preparations of the Pharmaceutical Dosage Forms and Real Samples

Spectracef^®^ pediatric sachets containing 50 mg of CFT were weighed and crushed to make them homogeneous in a mortar. To prepare a 1.0 mM stock solution, an equal amount of this powder was weighed and brought to the required volume with acetonitrile, sonicated for 15 min, and filtered. The electrochemical analysis was carried out from the stock solution prepared using the 0.1 M H_2_SO_4_ solution and keeping the acetonitrile amount (20%) constant. Recovery studies using the AdSDPV methods and the influence of excipients in pharmaceutical forms were researched for accuracy measurements.

To prepare a stock drug-free human serum solution with 0.1 mM CFT, 3.6 mL of commercial serum, 5.4 mL of acetonitrile, and 1 mL of 1 mM CFT were taken and accommodated in a 10 mL centrifuge tube. First, it was vortexed for 5 min and then centrifuged at 5000 rpm for 20 min. The supernatant was attentively transferred to a different clean tube. In this pre-process, acetonitrile acted as a precipitating agent for the protein residues. Subsequently, the analysis was undertaken as indicated in the basic analytical strategy. A calibration plot for the human serum samples was prepared to utilize the supporting electrolytes, which assisted with the recovery experiment. To control the accuracy of the designed nanosensor, recovery experiments were also applied to commercial deproteinated human serum samples using the standard addition method. All the analyses was saved with at least three repeatable measurements.

## 3. Results

### 3.1. Surface Morphological Characterization of a MWCNT/Chitosan NCs/Fe_2_O_3_-Modified GCE

The morphological characteristics of the proposed nanosensor were examined by scanning electron microscopy (SEM) and SEM energy-dispersive spectrometry (SEM-EDX). The surface morphologies of the bare GCE (A), MWCNT (B), Fe_2_O_3_ (C), chitosan NCs (D), and MWCNT/chitosan NCs/Fe_2_O_3_ (E) were obtained by SEM and are shown in Figure 2, along with the EDX spectra of MWCNT/chitosan NCs/Fe_2_O_3_ (F). As seen in Figure 2A, the bare GCE morphologies had a dark and smooth surface. The MWCNT had a filamentous structure, as shown in Figure 2B. As observed, the Fe_2_O_3_ nanoparticles were nearly spherical along with a degree of aggregation (Figure 2C). Conversely, as shown in Figure 2D, the morphology of chitosan NCs had irregularly shaped particles with a vast dispersion size. When the surface morphology of MWCNT/chitosan NCs/Fe_2_O_3_ was examined, Fe_2_O_3_ NPs were wrapped with MWCNT filaments, as seen in Figure 2E. For this reason, the electroactive surface area of the glassy carbon electrode was extended with the MWCNT/chitosan NCs/Fe_2_O_3_ modification agent. The MWCNT/chitosan NCs/Fe_2_O_3_ increased the signal answer of CFT and the conductivity of the bare GCE. The presence of Fe_2_O_3_ on the electrode surface confirmed the EDX spectra shown in Figure 2F. 

### 3.2. Electrochemical Behavior of CFT on the GCE

Responses of 1 × 10^−5^ M CFT were recorded using a bare GCE; MWCNT/chitosan NCs/Fe_2_O_3_ modified the GCE with the DPV technique, which is extremely sensitive. It was found that the MWCNT/chitosan NCs/Fe_2_O_3_ nanosensor significantly affected the electro-oxidation process compared with the bare electrode when the DPV results were analyzed (Figure 3). Moreover, the oxidation peak current of CFT with MWCNT/chitosan NCs/Fe_2_O_3_/GCE was 2.5 times higher than the anodic peak current of the bare GCE. Moreover, MWCNT/chitosan NCs/Fe_2_O_3_/GCE increased the signal of CFT by accelerating the electron transfer on the GCE surface.

### 3.3. Effect of the Scan Rate

Scan rate studies were used to evaluate whether the electrochemical process occurred under adsorption-controlled or diffusion-controlled processes and to inform us about the electrochemical oxidation mechanism. The effect of various scan rates in the range of 0.005–1.0 V s^−1^ on the potentials of CFT and the anodic peak currents were analyzed by the CV technique (Appendix A). The effect of the square root of ν (ν^1/2^) on the anodic peak current (I_p_) values of 1 × 10^−4^ M CFT in the 0.1 M H_2_SO_4_ solution for the designed nanosensor was researched. It was seen that with an increase in the scanning speed, the peak current increased and the peak potentials shifted to more positive values. The positive exchange in the peak potentials with the scan rate corresponded with the irreversible electron transfer processes. The relationship between the logarithm of the scan rate versus the logarithms of the anodic peak currents confirmed that this electrochemical determination was controlled by diffusion under adsorption-controlled processes for the MWCNT/chitosan NCs/Fe_2_O_3_ modified GCE. 

As seen in the equations below and the graphs (Appendix A), I_p_ was linear in the range of 0.005–1.0 V s^−1^ for the proposed nanosensor versus the square root of the scan rate:I_p_ (μA) = 0.010 *v* (mV s^−1^) + 0.749 (r = 0.982; *n* = 10) for MWCNT/chitosan NCs/Fe_2_O_3_/GCE in the 0.1 M H_2_SO_4_ solution;I_p_ (μA) = 0.344 *v*^1/2^ (mV s^−1^) − 1.055 (r = 0.993; *n* = 10) for MWCNT/chitosan NCs/Fe_2_O_3_/GCE in the 0.1 M H_2_SO_4_ solution;Log I_p_ (μA) = 0.725 log *v* (mV s^−1^) − 1.156 (r = 0.999; *n* = 10) for MWCNT/chitosan NCs/Fe_2_O_3_/GCE in the 0.1 M H_2_SO_4_ solution.

The electron transfer mechanism was indicated by the slope value of the linear equation between the log I_p_ and the log *v*. A slope value close to 1 showed an adsorption-controlled process whereas a slope of 0.5 implied a diffusion-controlled process [23]. The slope value of the linear equation between the log Ip and the log v was 0.72 for this nanosensor. This value was between the theoretical value of 1 and 0.5. Therefore, the mechanism was found; diffusion under adsorption-controlled processes.

### 3.4. Effect of pH on the Supporting Electrolyte

The effect of the supporting electrolyte on the pH was researched using the MWCNT/chitosan NCs/Fe_2_O_3_-modified GCE for the determination of CFT. The effect of the pH on CFT oxidation for supply optimum experimental conditions was examined in a range between 0.3 and 12.0 using an H_2_SO_4_ solution, a phosphate buffer, a BR buffer, and an acetate buffer with a constant amount of acetonitrile (20%) using the DPV technique. Furthermore, the effect of the pH value on the peak potential (E_p_) and peak current (I_p_) for 1 × 10^−5^ M was evaluated for MWCNT/chitosan NCs/Fe_2_O_3_/GCE. This nanosensor exhibited the highest peak current value and produced a well-defined peak shape with the 0.1 M H_2_SO_4_ solution (Figure 4A,B). Looking at Figure 4A, it can be seen that the E_p_ and I_p_ values of CFT decreased with an increasing pH. Hence, the 0.1 M H_2_SO_4_ solution was selected as the working medium for studying this nanosensor.

E_p_ (mV) = −44.0 pH + 944 (r = 0.983) (pH: 0.3 and 12.0).

The slope value of 44.00 mV per pH was close to the theoretical value of 59 mV, which demonstrated that the system obeyed the Nernstian behavior and involved an equal number of electrons and protons in the rate-determining step of the oxidation process [24,25].

Moreover, the slope of the straight line obtained from plotting the potential of anodic peak(Epa) versus the log ν was equal to 2.303 × $\frac{RT}{2\left(1-\alpha \right)nF}$ (Laviron equation), where *n* is the number of electrons transferred, R is the gas constant (8.314 J/mol/K), T is the temperature (298 °K), and F is Faraday’s constant (96,485 C/mol) [3]. The α value of 0.5 was calculated by assuming that one electron was involved in the rate-determining step of CFT oxidation. In this study, the *n* value was 1.49 (~1.0) for oxidation on the GCE. The values for *n* showed us the transfer of the same number of electrons and protons in the oxidation of the GCE.

### 3.5. Optimization of the Experimental Conditions for the Preparation of the Electrodes

#### 3.5.1. Optimization of the Amount of the MWCNT/Chitosan NCs/Fe_2_O_3_ Suspension

For the assay of CFT from the DPV results, it was approved that the MWCNT/chitosan NCs/Fe_2_O_3_-modified GCE was appropriate for developing a good perceiving platform. All the techniques, sequence changing, and mixing were tried. Therefore, the MWCNT/chitosan NCs/Fe_2_O_3_-modified electrode was used for further work. Later, the selection of the modifying agents made after a significant step, which was the optimization of the amounts of modifying agents on the surface of the electrode. The electron transfer kinetics of the porous film would be affected by the film thickness of the modifier on the electrode surface. The effect of the supply of the MWCNT/chitosan NCs/Fe_2_O_3_ suspension as a modifier on the surface of the electrode was researched using various amounts of the MWCNT/chitosan NCs/Fe_2_O_3_ mix between 1 and 5 μL with the DPV technique answer of CFT. Due to the high suspension density, 2.5 µL did not dry and fell on the surface as a layer. Therefore, the amount of the MWCNT/chitosan NCs/Fe_2_O_3_ suspension was kept constant at 1 µL.

#### 3.5.2. Effect of the Accumulation Time and the Potential for the GCE

The scan rate results indicated that the electrochemical reaction of CFT was adsorption-controlled on MWCNT/chitosan NCs/Fe_2_O_3_/GCE. Hence, the accumulation potential (E_acc_) and accumulation time (t_acc_), two significant parameters of AdSDPV, were appraised to increase the sensitivity. The effect of E_acc_ on the oxidation peaks of CFT was researched by AdSDPV between 0.0 V and 1.0 V. The CFT signal was watched with a changing accumulation potential between 0 and 1.0 V. The accumulation time was kept constant at 60 s in the stirred solution. The effect of the accumulation time on the signal of CFT was investigated in the range of 0–180 s by AdSDPV. The curve trends augmented up to 150 s and decreased after that. Therefore, optimum E_acc_ and t_acc_ parameters were chosen for all subsequent measurements at 0.0 V and 150 s, respectively.

### 3.6. Probable Oxidation Mechanism for CFT

The CV technique is the most significant method for researching the electroactive nature of organic compounds. This technique also assists in brightening the metabolic behavior of pharmaceutical compounds in the biological matrix. The CV technique was used to analyze the electrochemical redox behavior of CFT. The possible electro-oxidation mechanism of CFT was studied with similar model compounds as well as studies relative to the similar electroactive section in the CFT molecule. CFT consists of beta-lactam, aminothiazole, carboxylic acid, and other molecules. Cefdinir, cefpodoxime, cephalexin, and ceftazidime model compounds have similar electroactive moieties to CFT. The CFT signals were compared by conducting CV at a 0.1 V s^−1^ scan rate using an unmodified glassy carbon electrode with the 0.1 M H_2_SO_4_ solution (Figure 5).

CFT anodic peaks were observed at approximately 1.02 V and 1.20 V. The comparative working of the chosen compounds was undertaken by CV on the unmodified GCE to define the oxidation process of CFT. The anodic peak currents of cefdinir, cefpodoxime, and ceftazidime were observed at 0.97, 1.02, and 1.00 V with 0.1 M H_2_SO_4_, respectively. The beta-lactam ring, a common structure, was observed in cefditoren and the model molecules of cefdinir, cefpodoxime, ceftazidime, and cephalexin. The absence of an aminothiazole group in cephalexin led us to believe that the oxidation process might have occurred through this group. However, anodic peak currents of cephalexin could not be observed. That is, it could be suggested that CFT oxidation irreversibly occurred at the aminothiazole ring. The oxidation mechanism proposed by Taşdemir in his study supported this suggestion (Figure 6).

The slope of this equation was close to the slope of the expected theoretical value according to the formula Ep = f (pH) = −0.059 m/*n* [24], where *n* and m are the number of electrons and protons, respectively, which suggested that the number of electrons transferred in the electrode reaction in this case was equal to the number of protons.

The equal of Ep = f (pH) was shown as 2.303 × $\frac{RT\delta }{nF}\;$, where δ is the number of protons involved in the electrode reaction, *n* is the number of electrons transferred, R is the gas constant (8.314 J/mol/K), T is the temperature (298 °K), and F is Faraday’s constant (96,485 C/mol) [25]. In this study, the δ/*n* values were calculated as 0.074 for oxidation on the GCE. These values for δ/*n* showed us the transfer of the same number of electrons and protons in the oxidation of the GCE.

### 3.7. Analytical Performance

AdSDPV was used to appraise the difference between the anodic peak currents and the CFT concentrations for the MWCNT/chitosan NCs/Fe_2_O_3_/GCE nanosensor. The linear concentration range for CFT was determined under optimized conditions with MWCNT/chitosan NCs/Fe_2_O_3_/GCE. Calibration curves were obtained in a linear range between 0.2 μM and 10 μM (r = 0.999) by the AdSDPV technique for the MWCNT/chitosan NCs/Fe_2_O_3_-modified electrode (Figure 7).

I_P_ (μA) = 140,628 C (M) − 0.0005 (R^2^ = 0.999) for MWCNT/chitosan NCs/Fe_2_O_3_/GCE in the 0.1 M H_2_SO_4_ solution.

The CFT determination in commercial deproteinated human serum as the biological sample was performed with the MWCNT/chitosan NCs/Fe_2_O_3_-modified GCE. Linear calibration graphs in the range of 0.1 to 1 μM were obtained (Figure 8).

I_P_ (μA) = 502,844 C (M) − 0.0244 (r = 0.998) for the MWCNT/chitosan NCs/Fe_2_O_3_-modified GCE in a commercial deproteinated human serum sample.

The validation parameters of the suggested method were evaluated according to the official guidelines and ICH [26,27,28]. The obtained results are shown in Table 1. First, the limit of quantification (LOQ) and limit of detection (LOD) were calculated using the following equations: LOQ = 10σ/m and LOD = 3σ/m, where m is the slope in the calibration equation and σ is the standard deviation of the peak current for the low concentration of CFT in the calibration curve.

The LOQ and LOD were 5.50 nM and 1.65 nM for the modified MWCNT/chitosan NCs/Fe_2_O_3_/GCE sensor in the 0.1 M H_2_SO_4_ solution (20% acetonitrile), respectively. When CFT was investigated in the commercial deproteinated human serum samples, the LOQ and LOD were obtained as 13.68 nM and 4.10 nM for the MWCNT/chitosan NCs/Fe_2_O_3_-modified glassy carbon electrode, respectively. Therefore, it could be deduced that our suggested nanosensor facilitated detecting and determining CFT in the reported serum levels.

The precision (repeatability and reproducibility) of the proposed method was elucidated by calculating the RSD% of the respective parameters. Therefore, the reproducibility of the suggested method was shown with low RSD% values (<5% in the biological samples and <2% in the supporting electrolytes). 

### 3.8. Quantitative Analysis of CFT in Pharmaceutical Dosage Form and the Human Serum Samples

To assess the validity, applicability, and accuracy of the proposed method, MWCNT/chitosan NCs/Fe_2_O_3_ was applied to the measurement of CFT in Spectracef^®^ dosage form and the commercial deproteinated human serum samples using the standard addition method, which is given in Section 2.6. The CFT quantity in the biological samples and the pharmaceutical dosage forms were found to be linear from the calibration plot. After, a known amount of a standard CFT solution was added to the pharmaceutical dosage form and biological samples. The related voltammograms were saved. Finally, the quantity of CFT in the spiked samples was established using the calibration equations and the recoveries were calculated with the rate of the amount added to the quantity found. The perfect recoveries (96.97−98.62%) demonstrate the reliability and feasibility of the suggested methods for the analysis of the pharmaceutical dosage form and the human serum samples (Table 2).

### 3.9. Interference Study

The solution containing CFT and different interfering molecules in the 0.1 M H_2_SO_4_ solution (20% acetonitrile) were prepared before the analysis. To ensure the selectivity of the developed method, the effects of uric acid, ascorbic acid, dopamine, glucose, paracetamol, and several ions (K^+^, Cl^−^, Mg^+^, SO_4_^2−^, Na^+^, and NO_3_^−^) were investigated with the MWCNT/chitosan NCs/Fe_2_O_3_/GCE nanosensor. Table 3 shows the effectiveness of the interfering agents on CFT with the MWCNT/chitosan NCs/Fe_2_O_3_/GCE sensor. Furthermore, the table shows that the interference agent and CFT mixture prepared in various ratios (1:1, 1:10, and 1:100) gave a relative error in the CFT signal. In other words, a minimal change was seen in the current answer of CFT.

## 4. Discussion

The validation parameters enhanced the quality of the improved methods to determine CFT in the biological samples and pharmaceutical dosage forms. In addition, the developed nanosensor showed excellent stability, repeatability, and reproducibility (Table 1).

The LOQ and LOD values for CFT obtained from the modified GCE with MWCNT/chitosan NCs/Fe_2_O_3_ were also lower than the LOQ and LOD values for CFT with an unmodified GCE, as reported by Taşdemir [3]. In conclusion, our suggested nanosensor could, with ease, detect and determine CFT at concentration levels lower than the concentration indicated in the literature for this linear range (Table 4).

## 5. Conclusions

Adsorptive stripping differential pulse and cyclic voltammetry methods were used for the reliable analysis of CFT in a 0.1 M H_2_SO_4_ solution with pharmaceutical dosage forms and commercial deproteinated human serum samples using the proposed nanosensor. In addition, similar model compounds were used to clarify the oxidation mechanism of CFT by cyclic voltammetry. The advantages of the proposed methods were that they were selective, time-saving, cheaper, fully validated, and accurate as well as their plainness of the reagents and the experiment.

The LOQ and LOD values of CFT obtained from the modified GCE with MWCNT/chitosan NCs/Fe_2_O_3_ were also lower than the LOQ and LOD values for CFT for an unmodified GCE, as reported by Taşdemir [3]. In conclusion, our suggested nanosensor could, with ease, determine CFT at LOD levels lower than the concentration indicated in the literature for this linear range. Furthermore, according to the recovery studies, it has been proven that the analysis of CFT was correctly undertaken without interfering with the excipients in tablet dosage forms with the developed method.

## Figures and Tables

**Figure 1 micromachines-13-01348-f001:**
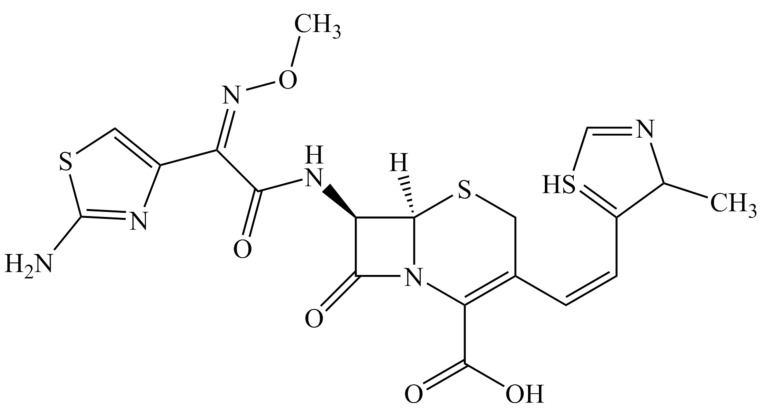
Molecular structure of cefditoren (CFT).

**Figure 2 micromachines-13-01348-f002:**
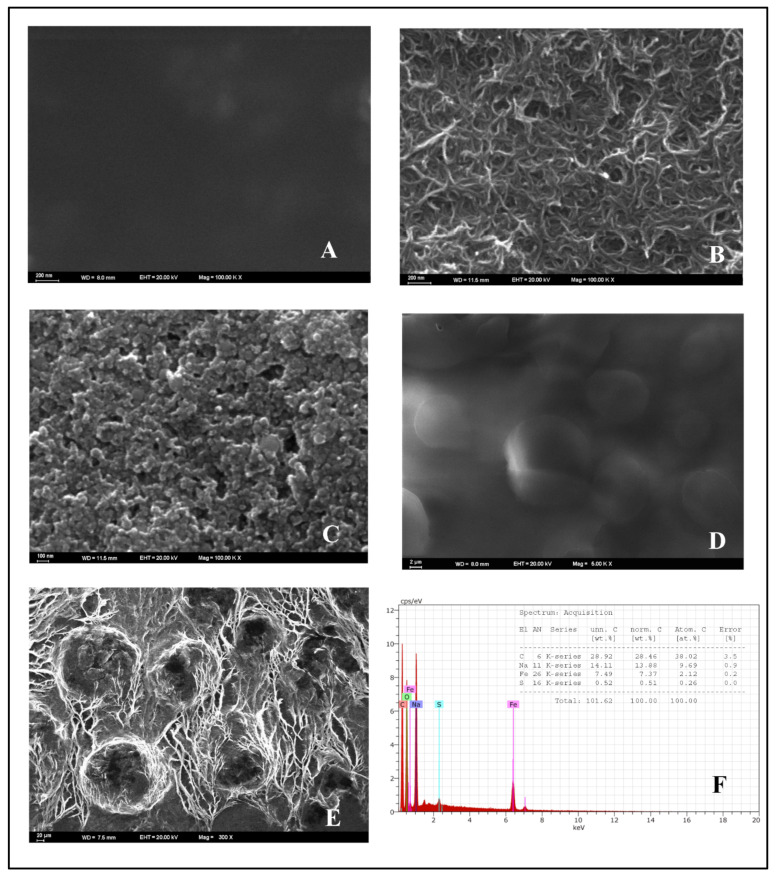
SEM images of (**A**) bare GCE, (**B**) MWCNT, (**C**) Fe_2_O_3_, (**D**) chitosan NCs, (**E**) MWCNT/chitosan NCs/Fe_2_O_3_, and (**F**) EDX spectra of MWCNT/chitosan NCs/Fe_2_O_3_.

**Figure 3 micromachines-13-01348-f003:**
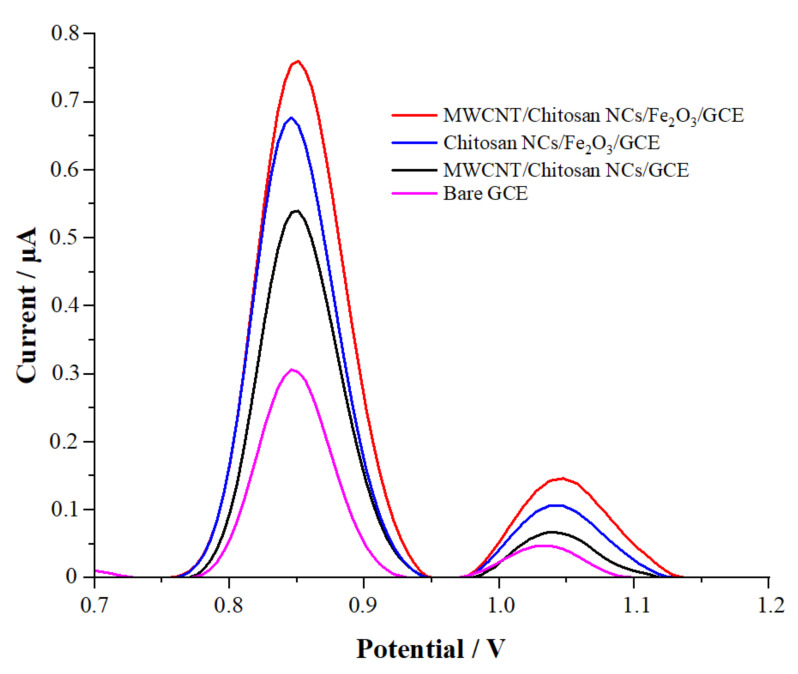
DP voltammograms of 1 *×* 10^−5^ M CFT solutions in pH 2.0 PBS (DPV parameters: 0 s equilibration time; 0.005 V step potential; 0.07 s pulse time; 0.02 V pulse potential; 0.02 V s^−^^1^ scan rate).

**Figure 4 micromachines-13-01348-f004:**
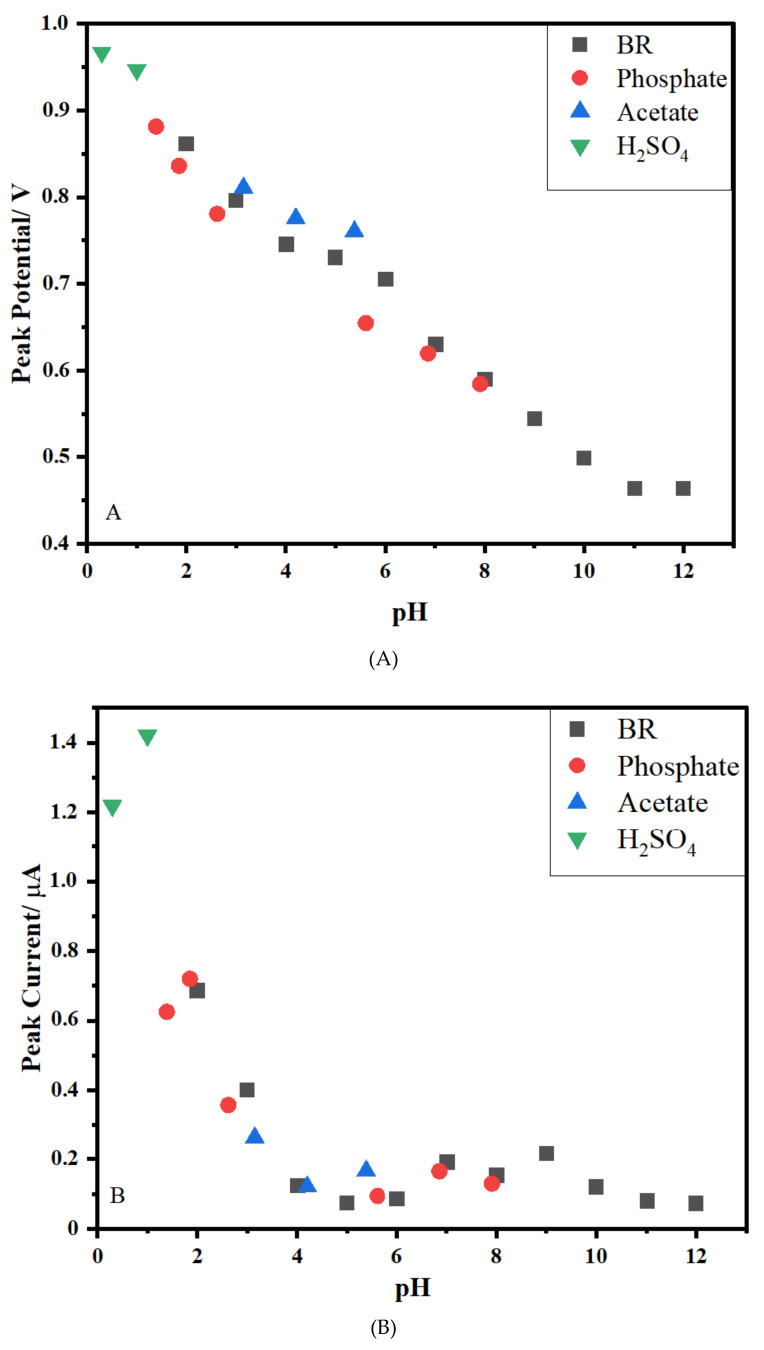
The plot of Ep (**A**) and Ip (**B**) of 1.0 × 10^−5^ mol L^−1^ CFT versus pH on MWCNT/chitosan NCs/Fe_2_O_3_/GCE (**A**,**B**) (■: BR buffer; ●: phosphate buffer; ▲: acetate buffer; ▼: H_2_SO_4_).

**Figure 5 micromachines-13-01348-f005:**
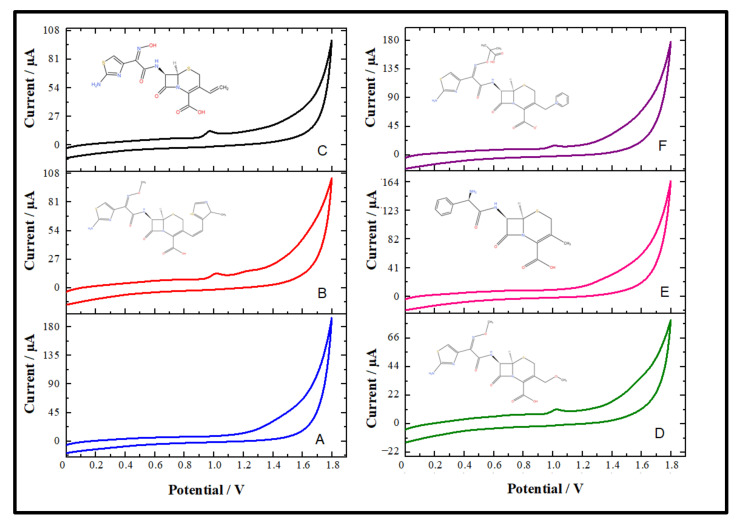
Cyclic voltammograms of a blank (blue line) (**A**), 1 × 10^−4^ M CFT (red line) (**B**), cefdinir (black line) (**C**), cefpodoxime (green line) (**D**), cephalexin (pink line) (**E**), and ceftazidime (purple line) (**F**) using a GCE in 0.1 M H_2_SO_4_ (CV parameters: 5 s equilibration time; 0 V begin potential; 0.005 V step potential; 0.1 V s^−1^ scan rate).

**Figure 6 micromachines-13-01348-f006:**
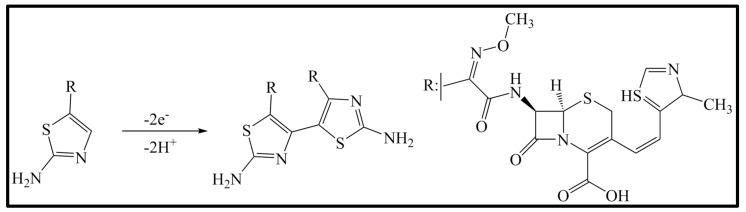
Proposed oxidation mechanism of CFT on a GCE. It is reprinted from Ref [26] in Science Direct.

**Figure 7 micromachines-13-01348-f007:**
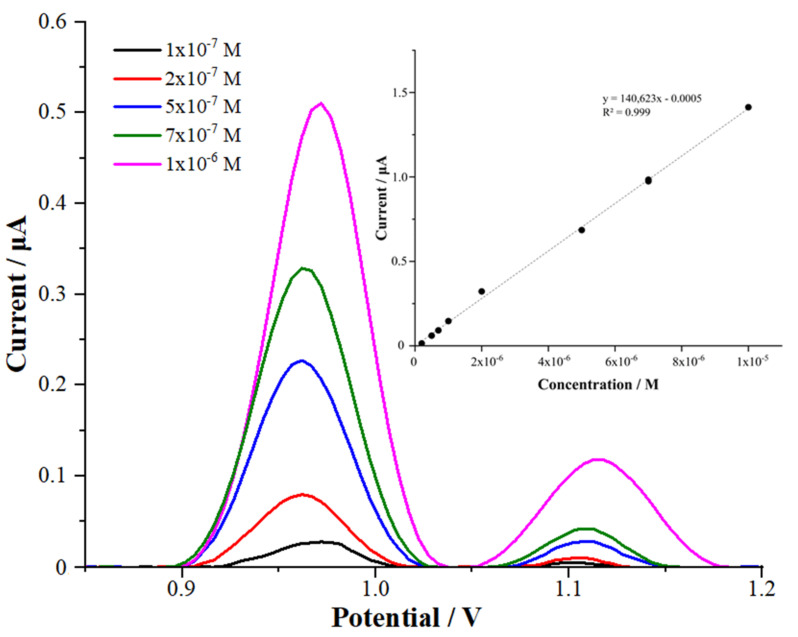
The AdSDP voltammograms of various concentrations of CFT in 0.1 M H_2_SO_4_ with MWCNT/chitosan NCs/Fe_2_O_3_/GCE (stripping provisos: accumulation time of 150 s and accumulation potential of 0.0 V).

**Figure 8 micromachines-13-01348-f008:**
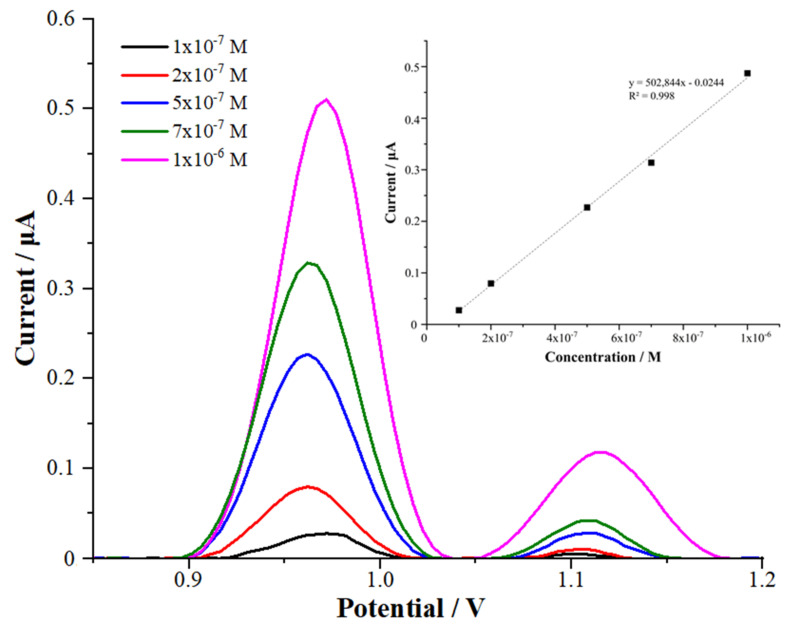
The AdSDP voltammograms of various concentrations of CFT in a commercial deproteinated human serum sample with MWCNT/chitosan NCs/Fe_2_O_3_/GCE (stripping provisos: accumulation time of 150 s and accumulation potential of 0.0 V).

**Table 1 micromachines-13-01348-t001:** Regression data of the calibration curves for the determination of CFT.

	CFT in 0.1 M H_2_SO_4_	CFT in Human Serum Samples
** *Linear range (μM)* **	0.2–10	0.1–1
** *Slope (μAM^−1^)* **	140,623	502,844
** *Standard error of the slope* **	2120.67	13,150.4
** *Intercept (μA)* **	−0.0005	−0.024
** *Standard error of the intercept* **	0.010	0.008
** *Correlation coefficient* **	0.999	0.998
** *LOD (nM)* **	1.650	4.100
** *LOQ (nM)* **	5.500	13.68
** *Intra-day repeatability (RSD %) ** **	0.711	0.929
** *Inter-day repeatability (RSD %) ** **	1.993	0.670
** *Reproducibility of electrodes within days (RSD %) ** **	0.710	1.540
** *Reproducibility of electrodes between days (RSD %) ** **	0.990	2.180

* Each value is an average of five measurements.

**Table 2 micromachines-13-01348-t002:** Recovery results obtained from the determination of CFT in pharmaceutical dosage form and human serum samples.

	MWCNT/Chitosan NCs/Fe_2_O_3_/GCE	
	Pharmaceutical Dosage Form (Spectracef^®^)	Serum Sample
** *Label amount (mg)* **	50	-
** *Found amount (mg)* **	48.42	-
** *RSD% ** **	1.78	-
** *Bias% ** **	3.26	-
** *Spiked amount (mg)* **	5.0	5.0
** *Found amount (mg) ** **	4.84	4.93
** *Average recovery (%) ** **	96.98	98.62
** *RSD% of recovery ** **	1.020	2.288
** *Bias% ** **	3.02	1.38

* Each value is the mean of five experiments.

**Table 3 micromachines-13-01348-t003:** Interference studies of CFT.

*Interference*		%
** *KCl* **	1:1	98.78
	1:10	99.69
	1:100	99.38
** *MgSO_4_* **	1:1	100.62
	1:10	100.13
	1:100	99.62
** *NaNO_3_* **	1:1	100.69
	1:10	100.74
	1:100	99.97
** *Glucose* **	1:1	99.96
	1:10	99.06
	1:100	98.62
** *Uric acid* **	1:1	100.67
	1:10	101.46
	1:100	104.67
** *Ascorbic acid* **	1:1	100.87
	1:10	99.85
	1:100	98.82
** *Paracetamol* **	1:1	100.65
	1:10	100.77
	1:100	99.81
** *Dopamine* **	1:1	100.54
	1:10	99.93
	1:100	99.19

**Table 4 micromachines-13-01348-t004:** Comparison of studies related to CFT detection.

*Electrode*	*Method*	*Linear Range (µM)*	*LOD (nM)*	*LOQ (nM)*	*Supporting Electrolyte*	*Peak Potential (V)*	*Ref*
**GCE**	AdSSWV	1.0–50.0	240	800	pH 4.0 (BRB)	+0.8	[3]
**HMDE**	AdSSWV	0.15–15.0	30	100	pH 6.0 (BRB)	−0.8	[3]
**Ion-Selective Electrodes**	Potentiometry	0.1–10,000	1.48	-	pH 7.0 (Borate buffer)	-	[15]
			50.1				
			50.1				
**MWCNT/Chitosan NCs/Fe_2_O_3_/GCE**	AdSDPV	0.2–10	1.65	5.50	0.1 M H_2_SO_4_	+ 0.8	This work

HMDE: hanging mercury drop electrode; GCE: glassy carbon electrode; DPV: differential pulse voltammetry; AdSSWV: adsorptive stripping squarewave voltammetry; MWCNT: multi-walled carbon nanotubes; PMA: phosphomolybdic acid; PTA: phosphotungstic acid; CTP: cefditoren pivoxil.

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
