# Peer review of "Rapid and Sensitive Electrochemical Assay of Cefditoren with MWCNT/Chitosan NCs/Fe2O3 as a Nanosensor"

_micromachines, 2022, doi:10.3390/mi13081348_

Round 1

Reviewer 1 Report

This paper presents the development of a MWCNT/ChitosanNCs/Fe2O3 modified GCE for Cefditoren determination. The paper has a logical structure and contains all classical aspects regarding the preparation, characterization and application of an electrochemical sensor. However, I have some suggestions which may help to improve the manuscript:

1. Fig. 1 is not clear. It must be sharper and the atoms should be written with larger fonts.

2. Line 38: Please replace "ion selective electrodes" with "potentiometry using ion-selective electrodes" 

3. The electrode used for pH-measurements should be indicated.

4. Line 119: "Different buffer solutions such as Britton-Robinson (BR) buffer (pH 2.0–12.0), sulphuric acid (pH 0.3–1.0) "

Please revise because sulfuric acid is not a buffer solution.

5. Line 135 and line 311: replace "%20 " with "20%"

6. Line 194: "versus the square root of . and scan rate (v1/2):" Please insert v1/2 after "square root of v" and not after "scan rate".

7. "The slope value of the linear equation between the log Ip and the log v was acquired as 0.72 for this nanosensor which is very close to the theoretical value of 1. This value proves the electron transfer mechanism of CFT that happens with adsorption-controlled. "

The slope of the log Ip vs. log v dependence is not closer to 1 than to 0.5 , it is in between. Therefore, the mechanism is mixt, both diffusion and adsorption controlled.

8. I suggest to discuss also the Ep=f(pH) dependence and to establish the ratio of protons and electrons involved in CFT electrooxidation at the MWCNT/Chitosan NCs/ Fe2O3 modified GCE .

9. I suggest to insert a figure with the overlayed CVs recorded at different scan rates.

10. As the authors developed a new modified GCE why did they use the bare GCE and not the modified one to study CFT electrooxidation mechanism?

11. The authors should specify that for the calibration graph they used the currents of the peak from the less positive values.

12. Line 284 and Table 1: The linear range for CFT in 0.1 M H2SO4 is given as 0.2-1 microM but in Fig. 2. the voltammograms and the calibration graph contain also values for concentrations up to 10-5 M and in the abstract the linear range is given up to 10 microM.

13. Table 1: Is the standard error of the slope not to high?

14. The authors showed that the developed modified GCE had a good repeatability. They should also mention how did they regenerate the electrode surface after analyte adsorption.

15. Table 4. First example (references 16) please check the LOD and LOQ values because it is not possible that the LOD is 8 times higher than the LOQ. 

Author Response

Dear Prof. Mr. Jeffery Zhou,

We would like to thank the Editor for considering our revised work entitled: “Rapid and Sensitive Electrochemical Assay of Cefditoren with MWCNT/Chitosan NCs/Fe2O3 as Nanosensor” by Goksu Ozcelikay, Nida Aydogdu, Sibel A. Ozkan for your consideration on possible publication in Micromachines’’.

We have replied to all the comments of the reviewers. All corrections were added to the text with yellow highlights. In this letter, bold parts belong to our answers.

We are looking forward to hearing from you.

Sincerely yours.

Sibel A. Ozkan

Author Response

(The authors gave the same response as above.)

Round 2

Reviewer 2 Report

Dear Editor,

The Authors are cleared almost all the queries raised by me, however, the above small minor corrections are required before publishing in the journal.

Author Response

Dear Prof. Mr. Scott Wang,

We would like to thank the Editor for considering our revised work. We have replied to the below comments of reviewer 2. All corrections were added to the text with yellow highlights in the supplementary material. In this letter, bold parts belong to our answers.

We are looking forward to hearing from you.

Sincerely yours.

Sibel A. Ozkan

  1. The author has used "r"values for the linear equations of scan rate studies in the manuscript and "R2”values were used in the supplementary material. Please use the same values in both cases.

Thank you very much for his/her suggestion. The "r"   value was used now in the supplementary material.

  1. Also, check the equation of log Ip vs log v (Manuscript: log Ip (µA) = 0.725 log v (mVs-1 ) - 1.156 (r = 0.999; n = 10), supplementary information: log Ip  = 0.7358 log υ – 1.1706 R² = 0.9979) and number of data points are not 10 but there are 9 in the figure SI2 (c)

Thank you very much for his/her suggestion. The equation of log Ip vs log v was checked and it was revised in the supplementary material. In the previous revision, one of the data points was lost because the equation box was in front of a point. But, it is revised now.
